# Fast Fourier Convolution

**Lu Chi**[1], **Borui Jiang**[2], **Yadong Mu**[1]*
[1]Wangxuan Institute of Computer Technology, [2]Center for Data Science
Peking University
{chilu,jbr,myd}@pku.edu.cn

## Abstract

Vanilla convolutions in modern deep networks are known to operate locally and at fixed scale (*e.g.*, the widely-adopted $3 \times 3$ kernels in image-oriented tasks). This causes low efficacy in connecting two distant locations in the network. In this work, we propose a novel convolutional operator dubbed as *fast Fourier convolution* (FFC), which has the main hallmarks of non-local receptive fields and cross-scale fusion within the convolutional unit. According to spectral convolution theorem in Fourier theory, point-wise update in the spectral domain globally affects all input features involved in Fourier transform, which sheds light on neural architectural design with non-local receptive field. Our proposed FFC is inspired to capsulate three different kinds of computations in a single operation unit: a local branch that conducts ordinary small-kernel convolution, a semi-global branch that processes spectrally stacked image patches, and a global branch that manipulates image-level spectrum. All branches complementarily address different scales. A multi-branch aggregation step is included in FFC for cross-scale fusion. FFC is a generic operator that can directly replace vanilla convolutions in a large body of existing networks, without any adjustments and with comparable complexity metrics (*e.g.*, FLOPs). We experimentally evaluate FFC in three major vision benchmarks (ImageNet for image recognition, Kinetics for video action recognition, MSCOCO for human keypoint detection). It consistently elevates accuracies in all above tasks by significant margins.

## 1 Introduction

Deep neural networks have been the prominent driving force for recent dramatic progress in several research domains. The goal of this paper is the exposition of a novel convolutional unit codenamed *fast Fourier convolution* (FFC). Motivating our design of FFC, we consider two desiderata. First, one of the core concepts in deep convolutional neural networks (CNNs) is *receptive field* that is deeply rooted in the visual cortex architecture. In convolutional networks, receptive field refers to the image part that is accessible by one filter. A majority of modern networks have adopted the architecture of deeply stacking many convolutions with small receptive field ($3 \times 3$ in ResNet [11] for images or $3 \times 3 \times 3$ in C3D [27] for videos). This still ensures that all image parts are visible to high layers, since stacking convolutional layers can increase the receptive field either linearly or exponentially (*e.g.*, using atrous convolutions [2]). However, for context-sensitive tasks such as human pose estimation, large receptive field in convolutions is highly desired. Recent endeavor on enlarging receptive field includes deformable convolution [9] and non-local neural networks [31].

Secondly, CNNs typically admit a chain-like topology. Neural layers provide different levels of feature abstraction. The idea of cross-scale fusion has celebrated its success in various scenarios. For example, one can tailor and send high-level semantics to shallower layers for guiding more accurate spatial detection, as shown in the seminal work of FPN [18]. Recent studies have considered

---

to reinforce cross-scale fusion in more complex patterns, as exemplified by HRNet [29] and Auto-DeepLab [19]. Our work is also partly inspired by GoogLeNet [26], which is among the early exploration of capturing and fusing multi-scale information in an operation unit, rather than among distant neural blocks.

We thus seek for a novel convolution operator that efficiently implements non-local receptive field and fuses multi-scale information. The key tool for our development is the spectral transform theory. In particular, we choose Fourier transform for incarnation, leaving further exploration of many other choices (*e.g.*, wavelet) as a future work. According to the spectral convolution theorem [15] in Fourier theory, updating a single value in the spectral domain globally affects all original data, which sheds light on design efficient neural architectures with non-local receptive field (*e.g.*, [34, 7]). In specific, we design a collection of operations with varying receptive fields, among which non-local ones are accomplished via Fourier transform. These operations are applied to disjoint subsets of feature channels. Updated feature maps across scales are eventually aggregated as the output.

To our best knowledge, FFC is the first work that explores an efficient ensemble of local and non-local receptive fields in a single unit. It can be used in a plug-and-play fashion for easily replacing vanilla convolutions in mainstream CNNs without any additional effort. In contrast, existing non-local operators can only be sparsely inserted into the network pipeline due to their expensive computational cost. FFC consumes comparable GFLOPs and parameters with respect to vanilla convolutions, yet conveys richer information. In the experiments, we apply FFC for tackling a variety of computer vision tasks, including image recognition on ImageNet, video action recognition on Kinetics dataset, and human keypoint detection on Microsoft COCO data. The reported performances consistently outstrip previous models by significant margins. We strongly believe that FFC can make inroads into domains of neural network design where uniform, local receptive field had previously reigned supreme.

## 2 Related Work

**Non-local neural networks**. The theory of *effective receptive field* [21] revealed that convolutions tend to contract to the central regions. This questions the necessity of large convolutional kernels. Besides, small-kernel convolutions are also favored in CNNs for mitigating the risk of over-fitting. Recently, researchers gradually realized that linking two arbitrary distant neurons in a layer is crucial for many context-sensitive tasks, such as classifying the action type in a spatio-temporal video tube or jointly inferring the precise locations of human keypoints. This is addressed by recent research on non-local networks. Early methods as in [31] rely on expensive self-convolutions, which incurs a series of follow-up research that seeks for acceleration (*e.g.*, [14]). Nonetheless, current paradigm of using non-local operators are sparsely inserting them into some network pipelines. The way that they can be densely knitted remains an unexplored research problem.

**Cross-scale fusion**. In CNNs, it is widely acknowledged that features extracted from different locations in a network are highly complementary, providing low-level (edges, blobs etc), mid-level (meaningful shapes) or high-level semantic abstraction. Cross-scale feature fusion has widely celebrated effectiveness in numerous ways. For example, FCN [20] directly concatenated feature maps of different scales, generating more accurate image segments. The visual object detection task requires both accurate localization and prediction of object categories. To this end, FPN [18] propagated features in a top-down manner, seamlessly bridging the high spatial resolution in lower layers and semantic discriminative ability in higher layers. Recently-proposed HRNet [29] conducted cross-scale fusion among multiple network branches that maintain different spatial resolutions.

**Spectral neural networks**. Recent years have witnessed increasing research enthusiasm on spectral neural networks. The spectral domain, previously harnessed only for accelerating convolutions, also provides a powerful building block for constructing deep networks. For example, [23] proposed spectral pooling that performs dimensionality reduction by truncating the representation in the frequency domain. [34] utilized wavelet based representation for restoring high-resolution images. [7] proposed paired spatial-spectral transforms and devised a number of new layers in the spectral domain. Our work advances above-mentioned research front via designing an operation unit that simultaneously uses spatial and spectral information for achieving mixed receptive fields.

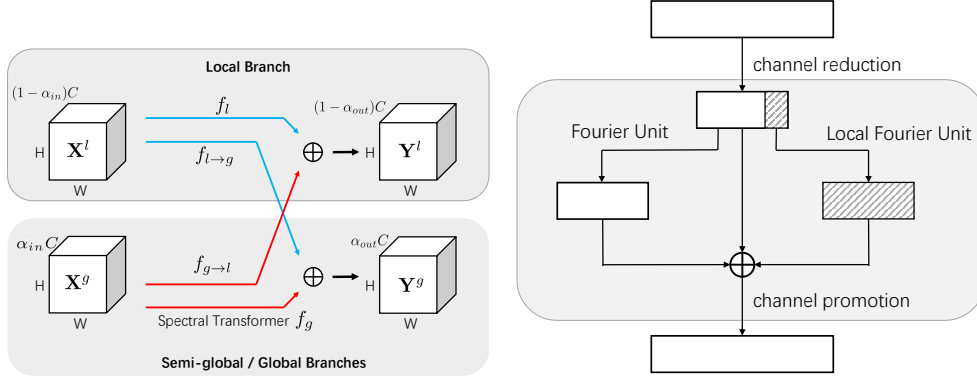

Figure 1: *Left*: Architecture design of Fast Fourier Convolution (FFC). "$\bigoplus$" denotes element-wise sum. Here $\alpha_{\text{in}} = \alpha_{\text{out}} = 0.5$. *Right*: Design of spectral transform $f_g$. See main text for more explanation.

# 3  Fast Fourier Convolution (FFC)

## 3.1  Architectural Design

The architecture of our proposed FFC is shown in Figure 1. Conceptually, FFC is comprised of two inter-connected paths: a spatial (or local) path that conducts ordinary convolutions on a part of input feature channels, and a spectral (or global) path that operates in the spectral domain. Each path can capture complementary information with different receptive field. Information exchange between these paths is performed internally.

Formally, let $\mathbf{X} \in \mathbb{R}^{H \times W \times C}$ be the input feature map of some FFC, where $H \times W, C$ represent the spatial resolution and the number of channels respectively. At the entry of FFC, we first split $\mathbf{X}$ along the dimension of feature channels, namely $\mathbf{X} = \{\mathbf{X}^l, \mathbf{X}^g\}$. The local part $\mathbf{X}^l \in \mathbb{R}^{H \times W \times (1-\alpha_{in})C}$ is expected to learn from local neighborhood and a second global part $\mathbf{X}^g \in \mathbb{R}^{H \times W \times \alpha_{in}C}$ is designed to capture long-range context. $\alpha_{in} \in [0, 1]$ represents the percentage of feature channels allocated to the global part. To simplify the network, assume the output is same sized to the input. Use $\mathbf{Y} \in \mathbb{R}^{H \times W \times C}$ for the output tensor. Likewise, let $\mathbf{Y} = \{\mathbf{Y}^l, \mathbf{Y}^g\}$ be a local-global split and the ratio of global part for output tensor is controlled by a hyper-parameter $\alpha_{out} \in [0, 1]$. The updating procedure within FFC can be described by following formulas:

$$\mathbf{Y}^l = \mathbf{Y}^{l \to l} + \mathbf{Y}^{g \to l} = f_l(\mathbf{X}^l) + f_{g \to l}(\mathbf{X}^g), \tag{1}$$

$$\mathbf{Y}^g = \mathbf{Y}^{g \to g} + \mathbf{Y}^{l \to g} = f_g(\mathbf{X}^g) + f_{l \to g}(\mathbf{X}^l). \tag{2}$$

For the component $\mathbf{Y}^{l \to l}$ which aims to capture small scale information, a regular convolution is adopted. Similarly, other two components ($\mathbf{Y}^{g \to l}$ / $\mathbf{Y}^{l \to g}$) obtained via inter-path transition are also implemented using regular convolutions to take full advantage of multi-scale receptive fields. Major complication stems from the calculation of $\mathbf{Y}^{g \to g}$. For statement clarity, we term $f_g$ as *spectral transformer*.

## 3.2  Implementation Details

**Spectral transformer.** The goal of global path in Figure 1 is to enlarge the receptive field of convolution to the full resolution of input feature map in an efficient way. We adopt discrete Fourier transform (DFT) for this purpose, using the accelerated version with Cooley-Tukey algorithm [8].

Figure 1 depicts our proposed spectral transformer. Inspired by the bottleneck block in ResNet, in order to reduce the computational cost, a $1 \times 1$ convolution is used at the beginning for halving the channels. Another $1 \times 1$ convolution is included to restore the feature channel dimension. As seen, between these two convolutions there are one Fourier Unit (FU) with global receptive field, a Local Fourier Unit (LFU) that is designed to capture semi-global information and operates on a quarter of feature channels, and a residual connection. The details of FU and LFU are given below.

```python
def FU(x):
    # x: input features with shape [N,C,H,W]

    # y_r / y_i is the real / imaginary part of the results of FFT, respectively
    y_r, y_i = FFT(x) # y_r/y_i: [N,C,H,⌊W/2⌋+1]
    y = Concatenate([y_r, y_i], dim=1) # [N,C*2,H,⌊W/2⌋+1]
    y = ReLU(BN(Conv(y))) # [N,C*2,H,⌊W/2⌋+1]
    y_r, y_i = Split(y, dim=1) # y_r/y_i: [N,C,H,⌊W/2⌋+1]
    z = iFFT(y_r, y_i) # [N,C,H,W]

    return z
```

Figure 2: Pseudocode of Fourier Unit (FU). The variables $N, C, H, W$ denote the sample number in a mini-batch, feature channels, image height and image width respectively.

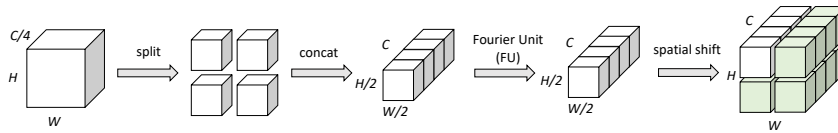

Figure 3: Illustration of the computational pipeline of Local Fourier Unit (LFU). To perform a valid addition with the output of FU (which has $C$ channels) in Figure 1, we spatially shift and replicate 4 copies of LFU's output (*i.e.*, in the outputted feature map, features in light green are copies of the white one).

**Fourier Unit (FU).** The purpose of FU is to first transform original spatial features into some spectral domain, conduct efficient global update on spectral data, and finally convert data back to the spatial format. Since Fourier transform manipulates complex numbers, it it crucial to ensure the input / output of FU are both real, such that it is compatible to other neural modules.

When applying 2-D Fast Fourier Transform (FFT) on some real signals, it is widely known that this brings an Hermitian matrix which is perfectly conjugate symmetric. Critically, applying inverse 2-D FFT operation on an Hermitian Matrix results in a matrix with only real elements [1]. The above properties can be utilized to ensure compatibility with other neural layers.

The pseudo-code of FU is shown in Figure 2. when FU is fed with real tensors, its results are conjugate symmetric. Without loss of useful information, we can thus retain only half of the results and trivially restore the other half by using conjugate symmetry. For ease of computation, we append the imaginary part to the real part, forming an additional dimension for the feature tensor. Essentially, the resultant tensor is treated as all reals. We can customize a series of new layers in the spectral domain. Following the practice in [7], a compilation of spectral 1x1 convolution, batch normalization and ReLU are conducted in our implementation of FU. Eventually the results are converted back to complex numbers by splitting them into real part and imaginary part along the auxiliary dimension. Inverse 2-D FFT operation creates an output tensor with all real numbers (*i.e.*, the variable $z$ in Figure 2).

**Local Fourier Unit (LFU).** FU manipulates the entire image. To capture and circulate semi-local information (*e.g.*, discriminative texture patterns in upper left of the input feature map), we further devise LFU, whose neural structure is shown in Figure 3. The key difference to FU lies in an additional split-and-concatenate step, which halves both of the spatial dimensions and renders four smaller feature maps. Standard FU is then applied to stacked feature maps. Inspired by the temporal shift in TSM [17], the results of FU go through a spatial shift and replication for fully restoring to the original resolution and channels.

LFU demands higher computational complexity compared with FU, mainly due to the increased channels. The effect of LFU varies with specific tasks. Section 4 clearly demonstrates that LFU complements FU in the scenario of image classification.

**Compatibility with vanilla convolutions.** A vanilla convolution can be adequately defined when specified from a few angles, including as kernel size, stride, channel grouping etc. Since $f_l, f_{g \to l}, f_{l \to g}$ all rely on regular convolution, they naturally inherit the property of regular convolutions. For spectral transformer $f_g$, there are two major considerations. First, as the convolution theorem reveals, large kernel size is not necessary for spectral transformer since any operation in spectral domain has a global receptive field. As a result the kernel size inside it is always fixed as 1.

Table 1: Parameter counts and FLOPs for vanilla convolution, separate component in FFC, and entire FFC respectively. $C_1$ and $C_2$ are the number of channels of input and output respectively. $H$ and $W$ collectively define the spatial resolution. $K$ is the convolutional kernel size. For clarity, here stride and padding are not considered. $\alpha_{in} = \alpha_{out} = \alpha$, where $\alpha$ is some parameter in $[0, 1]$.

| | #Params | FLOPs |
|---|---|---|
| vanilla | $C_1 C_2 K^2$ | $C_1 C_2 K^2 HW$ |
| $Y^{l \to l}$ | $(1-\alpha)^2 C_1 C_2 K^2$ | $(1-\alpha)^2 C_1 C_2 K^2 HW$ |
| $Y^{g \to g}$ | $\frac{\alpha^2}{2} C_2(C_1 + 3C_2)$ | $\frac{\alpha^2}{2} C_1 C_2 HW + \frac{13\alpha^2}{16} C_2^2 HW$ |
| $Y^{l \to g}$ | $\alpha(1-\alpha) C_1 C_2 K^2$ | $\alpha(1-\alpha) C_1 C_2 K^2 HW$ |
| $Y^{g \to l}$ | $\alpha(1-\alpha) C_1 C_2 K^2$ | $\alpha(1-\alpha) C_1 C_2 K^2 HW$ |
| FFC | $(1-\alpha^2) C_1 C_2 K^2 + \alpha^2 C_2(\frac{1}{2}C_1 + \frac{3}{2}C_2)$ | $(1-\alpha^2) C_1 C_2 K^2 HW + \alpha^2 C_2 HW(\frac{1}{2}C_1 + \frac{13}{16}C_2)$ |

Secondly, when mimicking the downsampling (*i.e.*, convolutional stride $> 1$) behavior of a vanilla convolution, inspired by OctConv [4], we use an average pooling before the channel-reducing step in Figure 1, which essentially does the job of downsampling with approximation.

The treatment of complex numbers in FU as previously stated ensures both input / output are real. FFC is therefore fully differentiable and can be inserted to most regular convolutional neural networks without additional modification.

Parameters $\alpha_{in}, \alpha_{out}$ can vary at different neural layers and control instant contribution from the semi-global and global branches. The shallowest layers are supposed to mainly exploit low-level local patterns, therefore we set $\alpha_{in} = 0$ in practice. The topmost layers highly demand contextual inference, which motivates the choice of $\alpha_{out} = 0, \alpha_{out} = 1$ (note that FFC boils down to vanilla convolution under $\alpha_{out} = 0$). In intermediate neural layers, we stick to $\alpha_{in} = \alpha_{out}$ without much empirical tuning.

**Extension to spatio-temporal video data.** FFC can be trivially extended to high dimensions by applying high-dimensional kernel (*i.e.* $t \times h \times w$) to all convolutional layers in FFC. The cornerstone for 3D-FFC is still 2D Fourier transform applied onto each individual feature channel.

### 3.3 Complexity analysis

Table 1 compares two major complexity metrics of FFC and vanilla convolution. Complexity of FFT / inverse FFT is omitted since they are parameter-free and their time complexities are negligible compared with other computation cost (*e.g.*, $\mathcal{O}(CHW \log(HW))$ in FU). As seen, FFC admits comparable cost with respect to vanilla convolution. Critically, FFC embodies its superiority when the large kernel size is needed, since the spectral transformer $f_g$ can still learn with global receptive field using $1 \times 1$ kernel size. Section 4 will provide more comparisons in real applications.

## 4 Experiments

We evaluate FFC on three visual tasks: image classification, video action classification and human keypoint detection. The main scope of the first study on ImageNet [16] is to investigate replacing vanilla convolutions in a number of modern CNNs using FFC, proving its versatility at different network architectures. A second experiment on the Kinetics video data is an investigation on high-dimensional data. The last task needs to precisely predict spatial locations of major human joints. This experiment is designed to prove that FFC can strike good balance between spatial specificity and global context.

### 4.1 Evaluation Protocols

**Image classification.** ImageNet [16] is widely adopted to pre-train network backbones for generalization to other more complex tasks. We validate FFC by replacing convolutions used in a variety of modern networks. Following typical settings in prior work, the input size of all the models is $224 \times 224$. Learning rate starts from 0.1 and decreases by a factor of 0.1 after 30, 60 and 80 epochs. Maximal training epochs are set to 90. Linear warm-up strategy is also adopted in the first 5 epochs. All the networks are optimized by SGD with a batch size of 256 on 4 GPUs. Common data

Table 2: **The top-1 accuracy of FFC under different ratios on ImageNet**. All models use ResNet-50 as their backbones. Note that $\alpha = 0$ is equal to using vanilla convolutions.

| Ratio | 0 | 0.125 | 0.25 | 0.5 | 0.75 | 1 |
|---|---|---|---|---|---|---|
| GFLOPs | 4.1 | 4.2 | 4.2 | 4.5 | 5.0 | 5.6 |
| #Params | 25.6 | 25.7 | 26.1 | 27.7 | 30.4 | 34.2 |
| Top-1 Accuracy | 76.3 | 77.3 | 77.6 | **77.8** | 77.6 | 75.2 |

Table 3: **Investigation of LFU on ImageNet**. ResNet-50 serves as the backbone for all.

| Ratio | LFU | GFLOPs | #Params | Top-1 Accuracy |
|---|---|---|---|---|
| 0.25 | | 4.2 | 26.1 | 77.6 |
| 0.25 | ✓ | 4.3 | 26.7 | **77.8** |
| 0.5 | | 4.5 | 27.7 | 77.8 |
| 0.5 | ✓ | 4.6 | 30.2 | **77.9** |

augmentation is utilized, such as scale jittering and random flipping. The validation accuracies are calculated in the same way as [11, 33, 12] based on $224 \times 224$ single center crop.

**Video action classification.** We choose Kinetics-400 as the testbed, which is a large-scale trimmed video dataset with more than 300K video clips and 400 categories in total. Following [31], ResNet-50 C2D and ResNet-50 I3D are selected as the backbones, with key convolutions replaced by 3D-FFC. All models use 8-frame video snippet with a temporal stride of 8, covering 64 frames in the temporal scale. Frame resolution is fixed as $224 \times 224$. All the models are initialized from the pretrained weights on ImageNet and trained on 4 GPUs with a batch size of 64 for total 100 epochs. The learning rate starts from 0.01 and decreases by a factor of 10 after 40 and 80 epochs. Dropout (0.5) after the global average pooling and weight decay (0.0001) are adopted to reduce over-fitting during training process. Identical data augmentation as [30, 31] is adopted, Following common practice in [24, 30], top-1 accuracy is reported by evenly sampling 25 clips per video with ten crops.

**Human keypoint detection.** The evaluations are fully conducted on Microsoft COCO keypoint benchmark (http://cocodataset.org). Recent sophisticated models (such as HRNet [29]) contain many distracting engineering tricks, thus not suitable for ablative study. Instead, we adopt SimpleBaseline [32] as the base model, and follow the experimental settings therein (*e.g.*, a spatial resolution of $256 \times 192$ or $384 \times 288$). For fairness, we also use the same person detector with detection AP 56.4 for the person category on COCO val2017 and several other treatments during inference. For example, averaging the heatmaps of the original and flipped images to get the final heatmap, and adjusting each keypoint's location via a quarter offset in the direction from the highest-value location to the second highest response.

### 4.2 Experiments on ImageNet

**Ablation studies on ratios** $\alpha_{in}, \alpha_{out}$**.** For intermediate layers, we set $\alpha_{in} = \alpha_{out} = \alpha$. The ratio $\alpha$ is varied from 0 to 1 to explore the best ratio between local and global paths based on ResNet-50 backbone. As shown in Table 2, two observations can be drawn. First, global path can bring a significant improvement to a model with only few additional GFLOPs or parameters. Secondly, the performance is not sensitive to $\alpha$. Large $\alpha$ implies more global operations and higher complexity, $\alpha = 0.5$ leads to a best accuracy, implying a good tradeoff. Considering the tradoff between performance and complexity, we set $\alpha = 0.25$ in the remaining experiments.

**Impact of cross-scale fusion.** FU and LFU differ in the scales that they operate. It is interesting to investigate how they complement one another. The ablation study is reported in Table 3. As seen, LFU consistently improves FU with moderate additional cost, although the accuracy elevation is marginal for this task.

Additionally, on ImageNet, using same parameters (*e.g.*, $\alpha = 0.25$), FFC with all cross-scale fusion achieves a top-1 accuracy of 77.6%. Removing global-to-local fusion or local-to-global fusion reduces the accuracy to 76.6%, 76.2% respectively. Removing $f_{l \to g}$ and $f_{g \to l}$ in Figure 1 only strikes an accuracy of 75.6%. This well validates the value of inter-path transitions.

**Plug-and-play with more CNNs.** We replace convolutions in more state-of-the-art networks using FFC to verify its generality. The results are shown in Table 4. As we can see, FFC can bring

Table 4: **Investigation of plugging FFC into more state-of-the-art networks on ImageNet.** The first two sets are top-1 accuracy scores obtained by various state-of-the-art methods, which we transcribe from the corresponding papers. Deeper models are listed in the second set. The last set reports the performances of plugging FFC into specific models (*e.g.*, FFC-ResNet-50 implies the use of a base model ResNet-50).

| Method | #Params | Top-1 Acc. |
|---|---|---|
| ResNet-50 [11] | 25.6 | 76.3 |
| SE-ResNet-50 [12] | 28.1 | 76.9 |
| $A^2$-Net [5] | - | 77.0 |
| Oct-ResNet-50 [4] | 25.6 | 77.3 |
| DenseNet-201 [13] | 20.0 | 77.4 |
| ResNeXt-50 (32 × 4d) [33] | 25.0 | 77.8 |
| Res2Net-50 (14w×8s) [10] | - | 78.1 |
| ResNet-101 [11] | 44.6 | 77.4 |
| ResNet-152 [11] | 60.2 | 78.3 |
| SE-ResNet-152 [12] | 67.2 | 78.4 |
| ResNeXt-101 (32 × 4d) [33] | 88.8 | 78.8 |
| AttentionNeXt-56 [28] | 31.9 | 78.8 |
| FFC-ResNet-50 | 26.7 | 77.8 |
| FFC-ResNext-50 (32×4d) | 28.0 | 78.0 |
| FFC-ResNet-101 | 46.1 | 78.8 |
| FFC-ResNet-152 | 62.6 | **78.9** |

Table 5: **Experimental results on Kinetics-400.** Three sets from top to bottom: recent state-of-the-art video models, our re-implemented base models, and models enhanced with FFC. All the models adopt ResNet-50 as backbones and read 8-frame input. "†" represents the model is finetuned with TSN framework [30].

| Method | GFLOPs | #Params | Top-1 |
|---|---|---|---|
| TSM [17] | 32.8 | 24.3 | 74.1 |
| $A^2$-Net [5] | 40.8 | - | 74.6 |
| Oct-I3D [4] | 25.6 | - | 74.6 |
| GloRe [6] | 28.9 | - | 75.1 |
| C2D | 19.6 | 24.3 | 71.9 |
| I3D | 28.4 | 28.4 | 72.6 |
| I3D + NL | 39.5 | 35.4 | 73.5 |
| C2D + NL | 30.7 | 31.7 | 73.8 |
| FFC-C2D | 20.2 | 24.9 | 73.5 |
| FFC-C2D + NL | 31.4 | 32.2 | 74.9 |
| FFC-I3D + NL | 40.2 | 35.9 | 75.1 |
| FFC-I3D + NL † | 40.2 | 35.9 | **76.1** |

significant improvement to various networks, even on powerful networks such as ResNeXt [33]. Surprisingly, FFC-ResNet-50 shows 0.4% better accuracy than ResNet-101 while costing only 60% parameters. The mixed, non-local receptive fields brought by FFC proves to significantly reduce the required depth for reaching a high level of accuracy. Additionally, FFC is also effective for deeper networks (+1.4% for ResNet-101 and +0.6% for ResNet-152), although these networks can achieve large receptive field by stacking many convolutiaonl layers, which shows that our method is complementary to traditional convolution.

## 4.3 Experiments of Video Classification

Training spatio-temporal deep models requires tremendous parameters and GPU memory for large video snippet. Limited by our GPU resources, here we only verify the proposed technique on ResNet-50 with 8-frame input. For fair comparisons, we only focus on comparing with recent state-of-the-art methods with the same backbone and input length. We also re-implement the popular models C2D, I3D and NL networks in [31] under the same settings as ours. All the results can be found in Table 5.

As can be seen, FFC can consistently improve the performance over baselines with only a few additional computations and parameters. For example, compared with C2D baseline, simply replacing some traditional convolutional layers with FFC can improve the accuracy from 71.9% to 73.5%, which is more effective and efficient than inflating 2D-convolution to 3D-convolution (achieving 0.9% higher accuracy, costing 29% less GFLOPs and 12% fewer parameters than I3D). Additionally, our method is complementary to the powerful non-local block [31]. Inserting non-local block to our models can further improve the performance. Comparing with other recent state-of-the-art methods,

Table 6: **Comparisons on the COCO val2017 dataset for human keypoint detection.** OHKM means Online Hard Keypoints Mining.

| Method | Backbone | Input Size | AP | AP$^{50}$ | AP$^{75}$ | AP$^M$ | AP$^L$ | AR |
|---|---|---|---|---|---|---|---|---|
| 8-stage Hourglass [22] | 8-stage Hourglass | $256 \times 192$ | 66.9 | - | - | - | - | - |
| CPN [3] | ResNet-50 | $256 \times 192$ | 68.6 | - | - | - | - | - |
| CPN + OHKM [3] | ResNet-50 | $256 \times 192$ | 69.4 | - | - | - | - | - |
| CPN + OHKM [3] | ResNet-50 | $384 \times 288$ | 71.6 | - | - | - | - | - |
| CSM + OHKM [25] | ResNet-50 | $384 \times 288$ | 73.8 | - | - | - | - | - |
| SimpleBaseline [32] | ResNet-50 | $256 \times 192$ | 70.4 | 88.6 | 78.3 | 67.1 | 77.2 | 76.3 |
| | | $384 \times 288$ | 72.2 | 89.3 | 78.9 | 68.1 | 79.7 | 77.6 |
| | ResNet-101 | $256 \times 192$ | 71.4 | 89.3 | 79.3 | 68.1 | 78.1 | 77.1 |
| | | $384 \times 288$ | 73.6 | 89.6 | 80.3 | 69.9 | 81.1 | 79.1 |
| | ResNet-152 | $256 \times 192$ | 72.0 | 89.3 | 79.8 | 68.7 | 78.9 | 77.8 |
| | | $384 \times 288$ | 74.3 | 89.6 | 81.1 | 70.5 | 81.6 | 79.7 |
| SRL-FFT [7] | ResNet-50 | $256 \times 192$ | 70.9 | 89.1 | 78.5 | 67.4 | 77.9 | 76.8 |
| | | $384 \times 288$ | 73.3 | 89.5 | 80.0 | 69.4 | 80.6 | 78.6 |
| | ResNet-101 | $256 \times 192$ | 71.8 | 89.3 | 79.6 | 68.4 | 78.7 | 77.6 |
| | | $384 \times 288$ | 74.3 | 90.1 | 81.3 | 70.5 | 81.5 | 79.7 |
| | ResNet-152 | $256 \times 192$ | 72.1 | 89.5 | 79.7 | 68.8 | 79.1 | 78.0 |
| | | $384 \times 288$ | 74.6 | 89.7 | 81.7 | 70.8 | 81.9 | 80.1 |
| FFC-SimpleBaseline | ResNet-50 | $256 \times 192$ | 71.8 | 89.3 | 79.5 | 68.5 | 78.7 | 77.7 |
| | | $384 \times 288$ | 73.9 | 89.5 | 80.6 | 70.2 | 81.3 | 79.4 |
| | ResNet-101 | $256 \times 192$ | 72.7 | 89.4 | 80.4 | 69.4 | 79.6 | 78.6 |
| | | $384 \times 288$ | 74.5 | 89.6 | 81.8 | 70.8 | 81.9 | 80.1 |
| | ResNet-152 | $256 \times 192$ | 72.9 | 89.6 | 80.5 | 69.6 | 79.9 | 78.7 |
| | | $384 \times 288$ | 74.8 | 89.6 | 82.2 | 71.1 | 82.1 | 80.3 |

our method is competitive. FFC-I3D with non-local blocks is able to achieve a new state-of-the-art result.

## 4.4 Experiments of Human Keypoint Detection on COCO

The task of human keypoint detection demands more fine-grained prediction compared with classification tasks in two other experiments, since finding human joints needs accurately localized feature semantics. It thus serves a good testbed for mixing local / non-local receptive fields.

All results measured in mean-average-precision (mAP) are in Table 6. Deeper models tend to suffer from saturated receptive fields, which offsets the benefits of plugging FFC. This explains that the most significant improvement is observed by ResNet-50, rather than deeper ones. As our second observation, larger spatial resolution of the input images arguably complicates the acquisition of large receptive field for standard CNNs. With FFC plugged in, much salient improvement can be seen compared with those cases with reduced resolution. Compared with the most recent model SRL-FFT [7], our method can achieve higher performance under all the experimental settings, which is mainly because our method can make full use of the local and global information. It should be noted that our method can achieve comparable performance with more deeper models. For example, under an input size of $256 \times 192$, FFC-ResNet-101 can perform better than SRL-FFT-ResNet-152.

## 5 Conclusion

We have proposed a novel convolutional operator dubbed as FFC. It harnesses the Fourier spectral theory for achieving non-local receptive fields in deep models. The proposed operator is also carefully designed to implement cross-scale fusion. Our comprehensive experiments on three representative computer vision tasks consistently exhibit large performance improvement that is clearly attributed to FFC. We strongly believe that FFC paves a new research front for designing non-local, scale-fused neural networks.

Acknowledgement: This work is supported by National Key R&D Program of China (2020AAA0104400), National Natural Science Foundation of China (61772037) and Beijing Natural Science Foundation (Z190001).

## 6 Broader Impact

Modern neural networks have evolved for decades, from the primary LeNet to recent Resnet, DenseNet etc. The deployment of neural network based models has greatly spurred the development of more industrial products, particularly for visual data oriented. Nonetheless, as our proposed FFC shows, a few key concepts in the architectural design of neural networks (such as receptive field) are still inadequately explored. This work presents a general technique FFC, with successful demonstrations in several crucial computer vision tasks, including image classification, video action recognition and human keypoint detection. A large body of context-sensitive computing tasks may benefit from FFC, as it can bring and fuse multi-scale neural receptive fields in a unified convolutional unit and thus help to capture richer contextual information.

Moreover, for deep learning research community, FFC may inspire more rethinking of neural network from a spectral aspect and lead to the development of more network backbones with boosted efficacy and accuracy. We believe that FFC has significant positive impact to both industry (particularly computer vision and natural language processing) and academia.

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
