[Reviews · NeurIPS 2020]

Review 1

Summary and Contributions: - Propose to use Fast fourier convolution to capture long range spatial dependencies and cross scale fusion in convolutional units - They have 3 different computations to capture local and global information - Motivation is to capture long-range/global information in early layers. - The authors build upon an idea to capture long-range information by operating in the spectral domain which has a global receptive field. They extend the previous method by incorporating cross-scale fusion and local information in the same block. - The operation is claimed to be generic with similar FLOPs as vanilla convolutions - Show improvements in Image recognition, Video recognition and Human keypoint detection.

Strengths: - Compared to the [Chi et al.2019] paper, their approach uses a standard spatial convolutions in addition to the spectral branch. This helps give better performance. Further, they incorporate multi-scale fusion to improve performance. - The improvement with a little to moderate increase in parameters is impressive. Further, the authors show results on 3 different tasks with improvements in all.

Weaknesses: - How useful were the inter-path transitions? An analysis on would help reason about how useful is this fusion. - In Table 5, how does the I3D + FFC compare with I3D + NL? - Analysis on how cross-scale fusion is helping the approach is necessary - The core component and methodology do not seem to be significantly different from [Chi et al.2019]. The approach seems to improve numbers through adding local convolutions and cross-scale fusion to the previous approach.

Correctness: - The empirical methodology is correct.

Clarity: - The paper is well written and easy to follow.

Relation to Prior Work: - The related work section is satisfactory. - Please add comparisons with "Attention Augmented Convolutional Networks" - ICCV'19.

Reproducibility: Yes

Additional Feedback: Post Rebuttal: I have read the other reviews and the rebuttal submitted by the authors. I think novelty is still an issue, especially given [Chi et. al]. I believe that the claim that it’s the first work to implement a single local + global unit is not correct. The AAC[R1] show a combination of self-attention with convolution in a single module. [R1] also reports lower overheads. The experiments performed by the authors show that the inter-path transitions seem to help but more intuition on this would be helpful. The observation during rebuttal that LFU is complementary to FU for all experiments and that of the method being complementary to NL is interesting. Based on the rebuttal and the other reviews, I will retain my score. [R1] Attention Augmented Convolutional Networks, ICCV’19


Review 2

Summary and Contributions: This paper proposes a novel convolution operator, which conduct convolution operations on spatial domain to capture local dependency and spectral domain to leverage non-local receptive field. It also introduces a semi-local operation to fuse multi-scale information. Experimental results on 3 large scale vision benchmark data sets show that the proposed method outperforms vanilla convolution at the cost of minor increasing of computation and params.

Strengths: 1. To the best of my knowledge, this paper should be the first to build a single conv unit which combines local and non-local information. 2. Adequate experimental results are given, which strongly show the advantages of the proposed method. 3. This work should be of wide interest to the NeurIPS community.

Weaknesses: Some part of this paper need to be clarified: 1. Why is vanilla conv used in l->g and g-l? Take l->g for an example, it introduces local info to global branch. I am just curious whether this transition is required. Can you just reserve l-l and g-g, and use channel shuffling on [Y^l, Y^g] to fuse local and global info, which should be more efficient? 2. Can you show some experimental results to show the necessity of l->g and g->l transitions? 3. For LFU, since the feature map patches are concatenated first and then a FU operator is applied, it also captures the information of whole feature map. Can you share some results of models with only LFU? 4. According to my understanding (correct me if I were wrong), since only 1x1 conv was used at spectral domain, it is equivalent to multiplying original feature map with a simple harmonic function at spatial domain, is it possible to do multiplication at spatial domain directly? The multiplied tensor should only requires several free parameters. 5. Could you clarify how to do channel splitting for group convolutions? Some groups for local operations and some for global ones, or every group will be split into local and global ones? --- I have read the rebuttal and my concerns are partially addressed. I am looking forward to further experimental results.

Correctness: To be best of my knowledge, the method and claims should be correct, but I have some concerns about the claims of LFU, which has been listed above.

Clarity: Yes, the writing of this paper is quite good.

Relation to Prior Work: Yes, relations to non-local network, spectral network and multi-scale feature fusion are clearly discussed.

Reproducibility: No

Additional Feedback: A typo in page 5 line 172 "Extension to spatiao-temporal..." --> "Extension to spatial-temporal..."


Review 3

Summary and Contributions: The paper proposes a convolution architecture that combines regular convolution (local receptive field) with a Fourier unit that takes the Fourier transform of the image (global receptive field). The goal is (i) to have non-local receptive field and (ii) to combine multi-scale info. Experiments on three different tasks/datasets (image recognition on ImageNet, video classification on Kinetics-400, keypoint detection on COCO) shows that this architecture consistently outperform baselines with similar number of parameters and FLOPs.

Strengths: 1. Strong experiment results on the proposed architecture. Across three different large-scale tasks/datasets (image recognition on ImageNet, video classification on Kinetics-400, keypoint detection on COCO), the proposed CNN architecture with Fourier unit improves on standard baselines. For example, it improves on standard ResNet on ImageNet by up to 1.5%. 2. Experiments are well-designed to explore the benefits of local and global context. The ablation experiment of local Fourier unit (Table 3) shows that this adds a small improvement. The ablation on Fourier unit (Table 2, alpha going from 0 to 1) illustrates that it's important to have both local context (regular convolution) and global context (with Fourier transform).

Weaknesses: 1. Lack of novelty: the idea of combining convolutions of multiple scales has been explored (e.g. Inception module, DenseNet, OctaveNet, etc.). The Fourier Unit seems identical to the design of [Chi et al. 2019]. The main contribution of this paper seems to be to combine the idea of Fourier unit in [Chi et al. 2019] with regular convolution. 2. The idea of multi-scale infomation is not validated. The experiments show that (near) global receptive field (i.e. with Fourier unit or local Fourier unit) is helpful. Local infomation (with regular convolution) is also helpful. However, it is not clear if information at intermediate scales is helpfu.

Correctness: The claims and empirical methodology appear to be correct.

Clarity: The paper is sufficiently clear and easy to follow.

Relation to Prior Work: Relation to prior work is clearly discussed. However, more detailed architecture constrast with [Chi et al. 2019] might better highlight the novelty of the paper. As written this difference is mentioned but not expanded on.

Reproducibility: Yes

Additional Feedback: What is the impact of the Fourier unit on memory and runtime during training/inference? ===== Update after authors' rebuttal: To clarify my comment regarding "intermediate scales" convolution: for an image of size n x n, a regular conv filter of size 5x5 will have a receptive field of 5x5, while a global Fourier-based filter will technically have a receptive field of n x n. The local Fourier unit in the paper cuts the image into size n/2 x n/2, so the receptive field there is n/2 x n/2. Intermediate scales would mean receptive field of size larger than 5x5 but smaller than n/2 x n/2.


Review 4

Summary and Contributions: This paper proposes a novel convolution operator named as Fast Fourier Convolution, which had the main hallmarks of non-local receptive fields and cross-scale fusion within the convolutional unit.

Strengths: The proposed Fast Fourier Convolution (FFC) is an interesting idea. It splits the spatial fields and reconstruct, then uses Fourier Transformer and Convolution to extract information from non-local fields. Moreover, FFC uses local branch and global branch to extract feature and fuse them for more information. And these operators increase few parameters and computation operations.

Weaknesses: The experimental results in this paper shows better performance when compared with the vanilla convolution. However, there are no performance comparison between the proposed FFC and other non-local methods and related convolutions such as dilated convolution. There is no ablation study (this is important because it can help us to understand the performance of each part of the proposed Fast Fourier Convolution).

Correctness: The claims and method are correct. The empirical methodology is also correct.

Clarity: Yes

Relation to Prior Work: Yes

Reproducibility: Yes

Additional Feedback:

[Author Response · NeurIPS 2020]

## To All Reviewers

**Main novelty**: Existing non-local models can only be sparsely inserted into the original network backbones, because either over-high complexity of the non-local operator (*e.g.*, [Wang et al.2018]) or the lack of multi-scale information (*e.g.*, [Chi et al.2019]). As pointed by R2, the proposed FFC is the first work that implements "a single conv unit which combines local and non-local information". Moreover, the complexity of FFC is comparable to vanilla convolution. These facts collectively enable FFC to directly replace vanilla convolutions in modern deep networks, achieving mixed receptive fields (local / semi-global / global) at each layer.

**Cross-scale fusion**: We would use empirical results to justify the necessity of cross-scale fusion (or inter-path transitions). For example, on ImageNet, using same parameters (*e.g.*, $\alpha = 0.25$), FFC with all cross-scale fusion achieves top-1 accuracy of 77.6%. Removing global-to-local fusion or local-to-global fusion reduces the accuracy to 76.6%, 76.2% respectively. Removing $f_{l \to g}$, $f_{g \to l}$ in Fig. 1 only strikes an accuracy of 75.6%. Similar observations are found on other benchmarks. Unfortunately these results were not included in the current draft due to our unwise page space organization. We will surely include the ablation studies in the revision.

## To R1

Our responses for the major concerns (difference with [Chi et al.2019], cross-scale fusion, and inter-path transitions) can be found in the Section "to all reviewers" of this rebuttal.

R1 requested "I3D + FFC v.s. I3D + NL". We are sorry that the experimental log of I3D + FFC is not successfully retrieved from our server. Nonetheless, for reference, Table 5 reports the accuracies of both C2D + FFC and C2D + NL, which are 73.5 v.s. 73.8. The corresponding GFLOPs are 20.2 v.s. 30.7. In comparison, the accuracy of original C2D is 71.9. We conclude that FFC and NL are similar in accuracy, but FFC is more efficient. Moreover, FFC / NL are complementary (FFC-C2D + NL -> 74.9).

Thanks for suggesting AA-ResNet (ICCV19). We will include its ResNet-50 results (77.7% v.s. FFC 77.8%) of as suggested in the revision.

## To R2

For questions 1 and 2, please refer to the Section "to all reviewers" of this rebuttal. The suggested "channel shuffling" essentially implements the same function to our current design (if we understand this suggestion correctly). However, its efficacy in comparison to FFC is unclear to us at this moment.

Table 3 investigates the final performance with or without LFU under different $\alpha$. It is observed that FU (global scale) / LFU (semi-global scale) are consistently complementary. We will conduct additional trials with only LFU as suggested in the revision.

R2 suggested "do multiplication at spatial domain directly". This applies to spectral 1x1 conv owing to the convolution theorem. However, it is not the case for spectral ReLU, which has a thresholding step (doing the job of frequency band-passing) that has no spatial correspondence.

FFC does channel splitting by the scheme of "Some groups for local operations and some for global ones". The resultant benefits are two-fold: it can be implemented by build-in group convolution in PyTorch. Moreover, the performance on CIFAR-100 using this scheme is slightly better than other alternatives, which serves as an indicator of better empirical choice.

## To R3

For the key novelty of the work and benefit of cross-scale fusion, please check Section "to all reviewers" of this rebuttal. We are not confident about the implication of "intermediate scales" as in the reviewing comment. FFC combines three scales: local (by vanilla convolution), semi-global (via local Fourier unit), and global (via Fourier unit). If "intermediate scales" was referring to the semi-global scale that operates on image patches, Table 3 investigates the effect of LFU.

## To R4

We indeed have included the requested comparisons in the submission, including non-local networks [Wang et al.2018], OctConv [Chen et al.2019a] and SRL [Chi et al.2019]. Dilated convolution is typically not chosen as a baseline in the literature of non-local models due to its inferior performance. Please see Tables 5 and 6 for more details. We fully agree with the reviewer that ablation studies are crucial for understanding each part of FFC. Please refer to Section "to all reviewers" of this rebuttal, where we provide detailed experimental results on ImageNet.

[Meta-Review · NeurIPS 2020]

Reviewers opinion was split on this paper. I am slightly less concerned with issues of novelty than some reviewers are - compositional methods (things that combine existing approaches) are often criticised for novelty but the approaches are commonly non-obvious a priori, and in this field almost all things are compositional. I think it is sufficiently non-obvious that the combination in this paper would provide the performance benefit it does. Hence it is of interest to practitioners in the field. The authors should temper their claims a bit about what they are the first to do, as mentioned by a couple of reviewers, and the authors must find a way to include their ablation tests in the rebuttal into the main paper. I appreciate the work done in the rebuttal to address reviewers' concerns. In summary, I felt the reviewers who gave negative scores also made important positive points and that the reasons given for low scores were either countered in the rebuttal, or not sufficiently important to prevent publication here. No doubt some reviewers will disagree, but there is always going to be disagreement in academic discourse!